# Physical activity and social support are associated with quality of life in middle-aged women

Thao Thi Phuong Nguyen[1,2]*, Hai Thanh Phan[3], Thuc Minh Thi Vu[4], Phuc Quang Tran[5], Hieu Trung Do[6], Linh Gia Vu[1,2], Linh Phuong Doan[1,2], Huyen Phuc Do[7], Carl A. Latkin[8], Cyrus S. H. Ho[9,10], Roger C. M. Ho[11,12]

1 Institute for Global Health Innovations, Duy Tan University, Da Nang, Vietnam, 2 Faculty of Medicine, Duy Tan University, Da Nang, Vietnam, 3 Institute for Preventive Medicine and Public Health, Hanoi Medical University, Hanoi, Vietnam, 4 Institute of Health Economics and Technology, Hanoi, Vietnam, 5 Hai Phong Medical University, Hai Phong, Vietnam, 6 Hung Yen Medical College, Hung Yen, Vietnam, 7 Center of Excellence in Evidence-based Medicine, Nguyen Tat Thanh University, Ho Chi Minh City, Vietnam, 8 Bloomberg School of Public Health, Johns Hopkins University, Baltimore, MD, United States of America, 9 Department of Psychological Medicine, National University Hospital, Singapore, Singapore, 10 Department of Psychological Medicine, National University Health System, Singapore, Singapore, 11 Department of Psychological Medicine, Yong Loo Lin School of Medicine, National University of Singapore, Singapore, Singapore, 12 Institute for Health Innovation and Technology (iHealthtech), National University of Singapore, Singapore, Singapore

* nguyentphuongthao73@duytan.edu.vn

**Data Availability Statement:** All relevant data are within the paper and its Supporting Information files.

**Funding:** The author(s) received no specific funding for this work.

## Abstract

### Purposes

This cross-sectional study assessed the quality of life and related factors of Vietnamese women during perimenopause in terms of vasomotor, psychosocial, physical, and sexual aspects.

### Materials and methods

A cross-sectional study on 400 middle-aged women was conducted in Hung Yen, a delta province in Vietnam. Data about socioeconomic characteristics, daily activity patterns, quality of life in terms of vasomotor, psychosocial, physical, and sexual aspects, and level of social support were collected. Tobit multivariate regression model was used to identify factors related to the quality of life among participants.

### Results

The symptoms of perimenopause appeared to worsen with the increase of age and the existence of such health issues as migraine and diabetes. Meanwhile, exercises, recreational activities, and social support appeared to alleviate the negative impact of perimenopausal symptoms on women.

### Conclusions

It is important to address the care needs of women during perimenopausal age, especially their sexual well-being, and development of specific healthcare services and programs

**Competing interests:** The authors have declared that no competing interests exist.

focusing on sport, entertainment, and support for women in perimenopause should be facilitated.

## Introduction

In perimenopause, the ovaries begin to degenerate and ovulation becomes irregular. This leads to the declination of sex hormones, such as estrogen and progesterone, and consequently, the appearance of the first clinical manifestations [1]. In addition to the common functional symptoms of perimenopause, including hot flashes, irregular periods, sleep disorders, vaginal dryness, decreased libido, the women are also facing the risk of cardiovascular disease, osteoporosis, and depression [1–5]. On the other hand, the psychological state of perimenopausal women is particularly sensitive and highly affected by interpersonal relations as well as socio-cultural factors [6, 7].

According to the General Statistics Office of Vietnam, the life expectancy of Vietnamese women is 76.2 years old, whilst perimenopausal symptoms occur from ages 40 to 58, and the average age of perimenopause is 51, meaning approximately one-third of the women's life is influenced by perimenopause, menopause, and post-menopause [2, 8]. The International Menopause Association document indicated that perimenopausal women who received proper care and support may prevent the risk of disease and increase the quality of life (QoL), compared to those who did not receive such care [9]. Perimenopausal age directly affects the women's health and QoL throughout the lifetime and given that the biological and psychological changes during perimenopause are strongly associated with the women's QoL. Hence, QoL of perimenopausal women needs careful consideration [10].

QoL predictors of women in the menopause transition and the influence of perimenopausal symptomatology on women's quality of life have been examined in several high-income countries, including the United States, China, Poland, and Spain [10–13]. While the association between perimenopause and QoL among middle-aged women has been documented in higher-income countries, there is a limited number of studies on this topic in low- and middle-income countries. Compared to the past, women have gained a greater foothold as well as played an increasingly important role in the development of the economy, and consequently, they have to cope with a much heavier workload [14]. On the other hand, in such Asian cultures or rural areas of low-and middle-income countries, a large number of women are still responsible for the majority of the domestic workload, which enhances the burden placed on women throughout their lifetime [15, 16]. Although the "double workload" usually leads to health deterioration in mid-life, research on the association between perimenopausal symptomatology and quality of life of middle-aged women remains limited in developing countries [17].

In addition to the onerous responsibilities, middle-aged women in Vietnam are also facing various barriers to access to health services as well as a lack of specific healthcare programs and interventions for promoting QoL of women [18–20]. Perimenopausal women in Vietnam thus require special attention not only because of their unique experiences but also due to the limitation in available research on their original vulnerability and additional health problems. Findings of research from previous studies of higher-income nations may not reflect the distinct socioeconomic and cultural characteristics surrounding health status and quality of life of middle-aged women in Vietnam, and which are unable to contribute to the development of effective interventions. Therefore, this study aims at assessing QoL and investigating the factors associated with QoL of women in perimenopause in a delta province in Vietnam.

## Materials and methods

### Study design, sample, and setting

A cross-sectional study was conducted in 4 wards of Hung Yen province from June to December 2018. Study subjects were women in Minh Khai, Hien Nam wards (representing urban areas) and Bao Khe, Lien Phuong wards (representing rural areas) who met the following eligibility criteria: (1) aged between 40 and 59; (2) living in Hung Yen city; (3) agreeing to participate in the study; (4) having no intellectual and cognitive issues; (5) identified as perimenopausal with an irregular period in the last 12 months; and (6) did not use hormone replacement during the 6 months before the study.

Four medical volunteers at ward health centers were recruited and involved in the current study. Selected interviewers were able to handle forms and keep track of paperwork. They were trained in face-to-face interviewing skills, using questionnaires, and locating the sampled study subjects at household. An interview guideline form was developed for collecting data and scoring the measure scales in questionnaires. Women in the wards were chosen using a simple random sampling method. A total of 400 women agreed to participate in this study. All participants were explained the objectives of the research and signed a written consent form.

### Measurements and instruments

Participants were directly interviewed to obtain information on socioeconomic characteristics, daily activity patterns, perimenopause-related QoL, and social support. Piloting the questionnaire was implemented on 20 middle-aged women to ensure that the questionnaire was logical, expressive, and understandable, and to avoid confusion and misunderstandings for research participants.

**Socioeconomic characteristics.** Information about age, marital status, occupation, living area, income quintiles, and monthly income was collected.

**Daily activity patterns.** The time of participants spent on work, housework, exercise, walking, and entertainment was measured in hours per day.

**Menopause-Specific Quality of Life (MENQOL) questionnaire.** QoL of perimenopausal women in this study was evaluated by the MENQOL questionnaire. MENQOL is composed of 4 domains, including vasomotor (items 1–3), psychosocial (items 4–10), physical (items 11–26), and sexual (items 27–29), with a total of 29 items on a Likert-scale format. Each item evaluates the impact of perimenopausal symptoms on QoL regarding the four aspects. Items pertaining to a specific symptom are checked as present or not present, and if present, the level of bothersome is assessed on a scale of zero (not bothersome) to six (extremely bothersome) [21, 22]. Means are calculated for each subscale by dividing the total score of the domain's items by the number of items within that domain. Items, i.e. perimenopausal symptoms, reported as not present are scored a "1", while those endorsed are regarded as "2," plus the number of the particular rating, thus the possible score on any item ranges from one to eight [22]. The score recorded is proportional to the decrease in QoL of perimenopausal women.

**Multidimensional Scale of Perceived Social Support (MSPSS).** The level of perceived support that the participants received was assessed using the Multidimensional Scale of Perceived Social Support (MSPSS). MSPSS is a brief assessment to measure a person's perceptions of support from Family, Friends, and a Significant Other. There are 4 items for each source of support; hence the scale consists of a total of 12 items. Each item can be scored from 1 (Very Strongly Disagree) to 7 (Very Strongly Agree) [23].

## Statistical analysis

The collected data were processed using the STATA version 14 (Stata Corp. LP, College Station, United States of America). Descriptive statistics were used to assess all of the questionnaire's items. The score of each item was converted ranging from 1 to 8 for analysis. 1 point is equivalent to participants answering 'No', while the scores from 2 to 8 correspond to the seven-point Likert scale (0–6). The variables of socioeconomic characteristics, daily activity, and perceived social support were considered as covariates, while the MENQOL (four domains) were served outcome variable. A t-test was used to describe the differences of perimenopause-related QoL between covariates. Tobit multivariate regression model was used to identify factors related to QoL of the participants, and $p < 0.05$ was regarded as statistically significant.

## Ethical approval

The protocol of this study was reviewed and approved by the Scientific Committee of Youth Research Institute, Ho Chi Minh Communist Youth Union (Code 177/QD/TWDTN-VNCTN).

## Results

A total of 400 middle-aged women participated in this study. The majority of respondents lived with their spouses (88.7%). The median age was 49.5 and the median of participants' income were USD 173.1. According to the MSPSS scale, the median of total score for perceived social support was 5.0 (**Table 1**).

As seen in **Table 2**, the prevalence of migraines and cardiovascular disease were 12.8% and 11.3%, respectively. Three-fifth of the sample used only one drug (58.0%). Almost all respondents used radio/television as a channel for health information (98.8%), second highest was local speakers (48.8%).

There were statistically significant differences between age groups in terms of psychological, physical, and sexual scores. The sexual scores of single women and women living with a spouse were 2.4 (SD = 1.7) and 2.9 (SD = 1.8), respectively. Physical scores were significantly different between urban and rural living areas (**Table 3**).

Factors associated with quality of life among Vietnamese women was shown in Table 4. Older age was related to higher MENQoL scores in all four domains. People living in the rural areas had a positive relationship with the score of sexual domains (Coef. = 1.40; 95%CI: 0.78; 2.01). Additionally, patients' health issues such as migraine, diabetes had higher scores in vasomotor, psychosocial, physical domains. Outpatient was positively associated with MENQoL in domains of psychosocial, physical, sexual.

In contrast, people being white-collar workers and businesses had lower scores of psychosocial and physical domains, while other occupations had higher scores of vasomotor and sexual domains. In addition, with higher entertainment active hours per day related to higher scores of all four domain, including vasomotor (Coef. = -1.51; 95%CI: -2.08; -0.93), psychosocial (Coef. = -0.96; 95%CI: -1.24; -0.68), physical (Coef. = -0.71; 95%CI: -0.86; -0.56), and sexual domains (Coef. = -1.19; 95%CI: -1.70; -0.69), similarly with doing exercise had positively related to the sexual domain (Coef. = -1.81; 95%CI: -2.52; -1.10). Participant was higher perceived social support score had negatively associated with vasomotor (Coef. = -0.54; 95%CI: -0.89; -0.18) and psychosocial domains (Coef. = -0.30; 95%CI: -0.48; -0.12).

## Discussion

In general, participants' quality of life in this study was most impaired in terms of sexual health. Age, living in rural areas, and the existence of such health issues as migraine and

**Table 1. Characteristics of participants.**

| Characteristics | n | % |
|---|---|---|
| **Total** | 400 | 100.0 |
| **Age group** | | |
| 40–44 | 106 | 26.5 |
| 45–49 | 94 | 23.5 |
| 50–54 | 95 | 23.8 |
| 55–60 | 105 | 26.3 |
| **Marital status** | | |
| Single/Divorced/Widow | 45 | 11.3 |
| Living with spouse | 354 | 88.7 |
| **Occupation** | | |
| Farmer | 168 | 42.0 |
| Blue-collar worker | 43 | 10.8 |
| White-collar worker | 41 | 10.3 |
| Business | 58 | 14.5 |
| Housemaker | 38 | 9.5 |
| Retirement | 32 | 8.0 |
| Others | 20 | 5.0 |
| **Living area** | | |
| Urban | 201 | 50.3 |
| Rural | 199 | 49.8 |
| **Income quintiles (million Vietnam Dong)** | | |
| Lowest (0.5–2) | 86 | 21.5 |
| Lower (2.5–4) | 121 | 30.25 |
| Medium (4.2–5) | 34 | 8.5 |
| High (5.3–12) | 79 | 19.75 |
| Higher (13–17.2) | 80 | 20 |
| | **Median** | **p25—p75** |
| **Age** | 49.5 | 44.0–55.0 |
| **Monthly income (USD)** | 173.1 | 129.8–354.8 |
| **Perceived Social Support (MSPSS)** | | |
| Significant other | 5.0 | 3.8–5.8 |
| Family | 5.3 | 4.8–6.0 |
| Friends | 4.8 | 3.8–5.5 |
| Total | 5.0 | 4.1–5.6 |
| **Active hours per day** | | |
| Work | 8.0 | 5.0–8.0 |
| Housework | 2.0 | 2.0–4.0 |
| Do exercise | 1.0 | 1.0–1.0 |
| Walk | 1.0 | 1.0–1.1 |
| Entertainment | 1.0 | 1.0–1.2 |

diabetes were positively related to increasing the symptoms of perimenopause. On a notable point, findings showed that women's quality of life in perimenopause might be improved by increasing the exercises, entertainment activities as well as perceived social support.

This result was lower than the previous mean MENQoL score which was explored in United Arab Emirates (3.03–3.61) [24], Saudi Arabia (2.28–3.19) [25], and China (2.33–2.84) [26] with sample sizes were respectively 70, 90 and 413. Furthermore, the present study

**Table 2. Health status, HRQOL, health service utilization, and health information sources.**

|  | n | % |
|---|---|---|
| **Health issues** |  |  |
| Migraine | 38 | 9.5 |
| Diabetes | 51 | 12.8 |
| Cardiovascular disease | 45 | 11.3 |
| **Have been attending inpatient** | 143 | 35.8 |
| **Have been attending outpatient** | 244 | 61.0 |
| **Poly drugs use** |  |  |
| None | 92 | 23.0 |
| Use one drug | 232 | 58.0 |
| Use more than one drug | 76 | 19.0 |
| **Health information sources** |  |  |
| Friends/relatives | 30 | 7.5 |
| Posters/banner | 4 | 1.0 |
| Internet | 56 | 14.0 |
| Text message | 14 | 3.5 |
| Radio/television | 395 | 98.8 |
| Local speakers | 195 | 48.8 |
| Newspapers/books | 61 | 15.3 |
| Medical staff | 41 | 10.3 |
| Social network | 10 | 2.5 |
|  | **Mean** | **SD** |
| **Number of inpatients** | 0.6 | 0.9 |
| **Number of outpatients** | 1.2 | 1.3 |

indicated the lowest mean score in the psychosocial domain, the finding was similar to the prior report [24]. In contrast, the sexual domains showed the highest mean score, which was consistent with the Chinese women group [26]. The quality of life of women during perimenopause is greatly affected by symptoms related to sexual health. Sex steroids play a critical role in the positive modulation of sexual behaviors throughout women's lifespan [27]. As the primary "female" hormone, estrogen promotes the growth and health of the female reproductive organs and keeps the vagina moisturized, elastic or stretchy, and well supplied with blood [28]. The decline in estrogen regulation during perimenopause is responsible for vaginal dryness and poor lubrication, sexual dysfunction, including orgasmic disorders, painful intercourse, loss of sexual interest, and other detrimental effects on the sexual well-being of women [1, 27, 28].

On the other hand, when women had more time for leisure and entertainment activities, quality of life regarding all four aspects achieved more remarkable improvement. A negative significant relationship between the number of exercise hours per day and the deterioration of QoL due to symptoms associated with sexual activity was observed. This finding is in line with previous research on the effects of exercises on the sexual function of women, in which exercise was pointed out to positively affect a variety of hormones, namely cortisol, estrogen, prolactin, oxytocin, and testosterone, and consequently, benefit physiological sexual arousal as well as alleviate anti-depressant-induced sexual dysfunction [29]. In addition to the beneficial impact on sexual symptoms, physical activities were also associated with fewer vasomotor, psychological, and somatic symptoms during menopause [30]. Meanwhile, social support (according to the MSPSS scale) and entertainment made a great contribution to the reduction of negative effects of perimenopausal symptoms on women's quality of life. Recreational

**Table 3. MENQOL domain scores in perimenopausal middle-age women according to different characteristics.**

| | Vasomotor | | | Psychosocial | | | Physical | | | Sexual | | |
|---|---|---|---|---|---|---|---|---|---|---|---|---|
| | Mean | SD | p-value | Mean | SD | p-value | Mean | SD | p-value | Mean | SD | p-value |
| **Total** | 2.2 | 1.7 | | 2.0 | 1.4 | | 2.2 | 1.1 | | 2.4 | 1.7 | |
| **Age group** | | | | | | | | | | | | |
| 40–44 | 2.0 | 1.7 | 0.05 | 2.0 | 1.3 | <0.01 | 1.9 | 1.0 | <0.01 | 2.1 | 1.7 | <0.01 |
| 45–49 | 2.2 | 1.8 | | 1.9 | 1.4 | | 2.2 | 1.2 | | 2.2 | 1.8 | |
| 50–54 | 2.6 | 2.0 | | 2.5 | 1.6 | | 2.4 | 1.1 | | 2.4 | 1.9 | |
| 55–60 | 1.9 | 1.4 | | 1.8 | 1.3 | | 2.2 | 0.9 | | 3.1 | 1.4 | |
| **Marital status** | | | | | | | | | | | | |
| Single/Divorced/Widow | 2.7 | 2.0 | 0.16 | 2.4 | 1.8 | 0.20 | 2.1 | 1.2 | 0.56 | 2.9 | 1.8 | 0.03 |
| Living with spouse | 2.1 | 1.7 | | 2.0 | 1.4 | | 2.2 | 1.0 | | 2.4 | 1.7 | |
| **Occupation** | | | | | | | | | | | | |
| Farmer | 1.9 | 1.6 | <0.01 | 1.9 | 1.2 | <0.01 | 2.0 | 1.0 | <0.01 | 1.9 | 1.4 | <0.01 |
| Blue-collar worker | 3.6 | 2.1 | | 3.0 | 1.7 | | 2.9 | 1.2 | | 3.1 | 2.3 | |
| White-collar worker | 2.6 | 1.8 | | 1.9 | 1.4 | | 2.0 | 0.9 | | 2.2 | 1.9 | |
| Business | 1.9 | 1.4 | | 1.6 | 1.3 | | 1.9 | 1.0 | | 2.9 | 1.6 | |
| Housemaker | 1.8 | 1.4 | | 1.9 | 1.5 | | 2.3 | 1.0 | | 3.1 | 1.3 | |
| Retirement | 1.7 | 0.8 | | 2.2 | 1.5 | | 2.4 | 0.9 | | 2.7 | 1.6 | |
| Others | 2.7 | 2.1 | | 2.3 | 1.7 | | 2.2 | 1.0 | | 3.2 | 1.9 | |
| **Living area** | | | | | | | | | | | | |
| Urban | 2.2 | 1.8 | 0.54 | 2.1 | 1.5 | 0.38 | 2.2 | 1.1 | 0.04 | 2.4 | 1.8 | 0.82 |
| Rural | 2.1 | 1.7 | | 2.0 | 1.4 | | 2.1 | 1.0 | | 2.4 | 1.6 | |
| **Income quintiles (million Vietnam Dong)** | | | | | | | | | | | | |
| Lowest (0.5–2) | 2.4 | 1.9 | 0.52 | 2.2 | 1.6 | <0.01 | 2.1 | 1.2 | 0.02 | 2.8 | 1.9 | 0.02 |
| Lower (2.5–4) | 2.4 | 1.9 | | 2.3 | 1.5 | | 2.4 | 1.1 | | 2.6 | 1.9 | |
| Medium (4.2–5) | 1.7 | 1.4 | | 2.1 | 1.3 | | 1.9 | 0.9 | | 2.3 | 1.8 | |
| High (5.3–12) | 1.9 | 1.2 | | 1.6 | 1.1 | | 2.0 | 0.8 | | 2.4 | 1.5 | |
| Higher (13–17.2) | 2.1 | 1.7 | | 1.9 | 1.2 | | 2.1 | 1.1 | | 1.9 | 1.4 | |

activities and the attention and support from people around them help perimenopausal women limit negative thoughts and guilt, minimize the risk of depression, hence positively impacting their QoL [31, 32].

Our study highlights the impact of sexual health on the quality of life among perimenopausal women. While sexual health during menopause transition is an important aspect to be addressed, sex life has remained a sensitive issue for a large number of middle-aged women in Vietnam, as can be observed from their hesitation when talking about sexuality. This psychological barrier has kept the women from sharing sex-related problems and seeking help to improve sexual symptoms during the menopause transition. Therefore, developing sources of information that are convenient for self-inform (websites, bulletin boards at health facilities, etc.) about methods to improve sexual-related symptoms during perimenopause may be a feasible solution in the current context of Vietnamese. In order to encourage perimenopausal women to participate in leisure activities and sport, programs to improve knowledge and health care advice, as well as specific guidelines on physical activities that are particularly beneficial for women at this stage, especially those having such conditions as migraine, diabetes, and cardiovascular disease, should be expanded. Additionally, it is important that women at perimenopausal age receive sufficient health care and emotional support from their family and friends.

In this study, we used the MENQOL questionnaire—the quality of life scale for perimenopausal women that has been standardized internationally, combined with a multi-stage

**Table 4. Associated factors with MENQOL domain scores.**

| | Vasomotor | | Psychosocial | | Physical | | Sexual | |
|---|---|---|---|---|---|---|---|---|
| | Coef. | 95% CI | Coef. | 95% CI | Coef. | 95% CI | Coef. | 95% CI |
| **Age** | 0.12*** | 0.06; 0.18 | 0.03** | 0.00; 0.06 | 0.05*** | 0.03; 0.06 | 0.17*** | 0.11; 0.22 |
| **Marital status** (Living with spouse vs Single/Divorced/Widow) | | | | | 0.36** | 0.05; 0.67 | | |
| **Occupation** (vs Farmer) | | | | | | | | |
| Blue-collar worker | 1.42*** | 0.34; 2.50 | | | 0.22 | -0.11; 0.56 | 1.23** | 0.21; 2.24 |
| White-collar worker | 0.54 | -0.51; 1.60 | -0.93*** | -1.53; -0.33 | -0.27* | -0.57; 0.03 | | |
| Business | | | -1.70*** | -2.33; -1.07 | -0.46*** | -0.73; -0.20 | 1.66*** | 0.86; 2.45 |
| Housemaker | -0.97 | -2.15; 0.20 | -0.71** | -1.37; -0.05 | | | 2.19*** | 1.27; 3.11 |
| Retirement | | | | | | | 1.07* | -0.00; 2.14 |
| Others | | | -0.52 | -1.29; 0.26 | | | 1.50** | 0.24; 2.77 |
| **Living area** (Rural vs Urban) | 0.52 | -0.16; 1.20 | | | 0.15 | -0.04; 0.34 | 1.40*** | 0.78; 2.01 |
| **Active hours per day** | | | | | | | | |
| Work | | | 0.12** | 0.02; 0.23 | | | | |
| Housework | | | | | -0.05 | -0.12; 0.01 | | |
| Do exercise | | | | | | | -1.81*** | -2.52; -1.10 |
| Walk | | | 0.42* | -0.07; 0.90 | | | | |
| Entertainment | -1.51*** | -2.08; -0.93 | -0.96*** | -1.24; -0.68 | -0.71*** | -0.86; -0.56 | -1.19*** | -1.70; -0.69 |
| **Perceived Social Support (MSPSS)** | -0.54*** | -0.89; -0.18 | -0.30*** | -0.48; -0.12 | -0.08 | -0.18; 0.03 | | |
| **Health issues** (Yes vs no) | | | | | | | | |
| Migraine | 1.24** | 0.06; 2.43 | 0.94*** | 0.31; 1.57 | 0.51*** | 0.18; 0.84 | | |
| Diabetes | 2.09*** | 0.96; 3.22 | | | 0.61*** | 0.29; 0.93 | | |
| Cardiovascular disease | 1.13** | 0.01; 2.25 | -0.59* | -1.19; 0.00 | | | | |
| **Number of inpatient** | -0.42* | -0.87; 0.02 | | | | | | |
| **Number of outpatient** | 0.26 | -0.07; 0.58 | 0.45*** | 0.31; 0.59 | 0.16*** | 0.07; 0.24 | 0.30*** | 0.09; 0.51 |

\*\*\* p<0.01

\*\* p<0.05

\* p<0.1.

sampling method, thus the above results may be applied to other delta areas in Vietnam. However, some limitations should be acknowledged. Since this research was designed in the form of a cross-sectional study, we were unable to establish a causal relationship between the factors and the outcomes of interest. Also, recall errors and social desirability bias may occur during the self-report process.

## Conclusions

Sexual well-being plays a major role in the quality of life of perimenopausal women. While health conditions such as migraine, diabetes, and cardiovascular disease enhance the symptoms of the menopause transition, the negative impact of these symptoms on women's quality of life can be improved by exercises, recreational activities, and social support. For middle-aged women in Vietnam to overcome menopause transition more easily, specific health care services for this population should be further developed and expanded.

## Supporting information

**S1 Dataset.**
(ZIP)

## Acknowledgments

We would like to express our gratitude to healthcare professionals and participants in Hung Yen city for supporting us perform this study.

## Author Contributions

**Conceptualization:** Thao Thi Phuong Nguyen, Hai Thanh Phan, Thuc Minh Thi Vu, Phuc Quang Tran, Hieu Trung Do, Linh Phuong Doan, Huyen Phuc Do, Carl A. Latkin, Cyrus S. H. Ho, Roger C. M. Ho.

**Data curation:** Thao Thi Phuong Nguyen, Phuc Quang Tran, Hieu Trung Do, Linh Gia Vu, Linh Phuong Doan.

**Formal analysis:** Thao Thi Phuong Nguyen, Linh Gia Vu, Linh Phuong Doan, Cyrus S. H. Ho.

**Investigation:** Thao Thi Phuong Nguyen, Hai Thanh Phan, Thuc Minh Thi Vu, Phuc Quang Tran, Hieu Trung Do, Linh Phuong Doan.

**Methodology:** Thao Thi Phuong Nguyen, Hai Thanh Phan, Thuc Minh Thi Vu, Phuc Quang Tran, Linh Gia Vu, Huyen Phuc Do, Carl A. Latkin, Cyrus S. H. Ho, Roger C. M. Ho.

**Project administration:** Thao Thi Phuong Nguyen, Hai Thanh Phan, Thuc Minh Thi Vu, Phuc Quang Tran, Hieu Trung Do, Linh Phuong Doan, Huyen Phuc Do, Cyrus S. H. Ho, Roger C. M. Ho.

**Resources:** Linh Gia Vu, Huyen Phuc Do, Carl A. Latkin.

**Software:** Linh Gia Vu.

**Supervision:** Thao Thi Phuong Nguyen, Thuc Minh Thi Vu, Phuc Quang Tran, Hieu Trung Do, Linh Phuong Doan, Huyen Phuc Do, Carl A. Latkin, Cyrus S. H. Ho, Roger C. M. Ho.

**Validation:** Thuc Minh Thi Vu, Linh Gia Vu, Carl A. Latkin, Cyrus S. H. Ho, Roger C. M. Ho.

**Visualization:** Hai Thanh Phan, Thuc Minh Thi Vu, Phuc Quang Tran, Huyen Phuc Do, Carl A. Latkin, Roger C. M. Ho.

**Writing – original draft:** Thao Thi Phuong Nguyen, Hai Thanh Phan, Linh Gia Vu, Huyen Phuc Do, Carl A. Latkin.

**Writing – review & editing:** Thao Thi Phuong Nguyen, Hai Thanh Phan, Thuc Minh Thi Vu, Phuc Quang Tran, Hieu Trung Do, Linh Phuong Doan, Huyen Phuc Do, Carl A. Latkin, Cyrus S. H. Ho, Roger C. M. Ho.

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
