## [Decision Letter · Decision Letter 0]

6 Oct 2021

PONE-D-21-21709Physical activity and social support are associated with quality of life in middle-aged womenPLOS ONE

Dear Dr. Nguyen,

Thank you for submitting your manuscript to PLOS ONE. After careful consideration, we feel that it has merit but does not fully meet PLOS ONE’s publication criteria as it currently stands. Therefore, we invite you to submit a revised version of the manuscript that addresses the points raised during the review process.

I apologize for the delay; there have been difficulties finding reviewers. Please note that the reviewers have both indicated that more explanation of your methods and findings are needed.

We look forward to receiving your revised manuscript.

Kind regards,

Rebecca Lee Smith, D.V.M., M.S., Ph.D.

Academic Editor

PLOS ONE

2. Please state whether you validated the questionnaire prior to testing on study participants. Please provide details regarding the validation group within the methods section.

Additional Editor Comments (if provided):

Reviewers' comments:

Reviewer's Responses to Questions

**Comments to the Author**

1. Is the manuscript technically sound, and do the data support the conclusions?

Reviewer #1: No

Reviewer #2: No

2. Has the statistical analysis been performed appropriately and rigorously? 

Reviewer #1: No

Reviewer #2: Yes

3. Have the authors made all data underlying the findings in their manuscript fully available?

Reviewer #1: No

Reviewer #2: No

4. Is the manuscript presented in an intelligible fashion and written in standard English?

Reviewer #1: No

Reviewer #2: Yes

5. Review Comments to the Author

Reviewer #1: While the author’s purpose for the manuscript is to contribute to the development of effective interventions for perimenopausal women in Vietnam using QOL as a health indicator, the manuscript needs major revisions to address statistical analysis and clarity of the results.

How were the income quintiles defined in terms of unit of measurement? Can you provide a range of these units in each quintile (i.e. Lowest (0-10,000 USD))?

Is the data available? If not, can you provide an explanation of its absence.

While the authors mention a few of the significant factors in table 2, they seem to miss discussing other significant variables. Can you please discuss table 2 a bit more in terms of other significant variables? Also, can you discuss the importance of the Total row in table 2?

I appreciate the information table 3 provides, however, the discussion of the table seems unclear.

Table 4 seems to provide coefficient values, however, the authors discuss odds ratios in their discussion of this table. Please choose either the coefficient values or ORs when reporting regressions.

I am unsure of why a tobit regression was used because the authors do not explain the presence of censored data. Can you please explain the importance of this model in your manuscript?

The reporting of the multivariate model is unconventional and should include results for each domain unless there is a valid reason for the absence of these results.

Please expound on the main results in the discussion. I also recommend including literature from the SWAN study and the Midlife Women’s Health Study for comparison in the discussion.

There are major grammatical errors needing to be addressed, I suggest language editing of the manuscript.

Reviewer #2: This is a very nicely designed study, but the description and conclusions need improvement.

Much of the needed methods description is missing. How many interviewers were involved, and how were they trained? Did they use a guided interview form for socioeconomic and activity data? Were interviews recorded and transcribed/scored later, or were they scored at the time of the interview? How were participants recruited? Please follow the STROBE guidelines (https://www.equator-network.org/reporting-guidelines/strobe/).

You are making causal claims with your language in the discussion, when your data are only correlational. Be careful about what you can conclude from a cross-sectional study design, and rewrite the description of your findings in terms of association only.

How much of the effect of exercise and leisure activity could be confounded by income and socio-economic status effects?

There are quite a few typos, and no line numbers have been provided. Please edit carefully and add line numbers for ease of review.

Specific comments:

p. 11: suggest using “Asian” instead of “oriental”

Table 1: What were the income quintile values? Also, can you add the MENQOL data here?

6. PLOS authors have the option to publish the peer review history of their article (what does this mean?). If published, this will include your full peer review and any attached files.

Reviewer #1: **Yes: **Brandi Patrice Smith

Reviewer #2: No

---

## [Author Response · Author response to Decision Letter 0]

12 Nov 2021

I/ Journal requirements

1. Please ensure that your manuscript meets PLOS ONE's style requirements, including those for file naming. The PLOS ONE style templates can be found at …

Responses:

- We have reviewed and adjusted the manuscript according to the PLOS ONE style templates

2. Please state whether you validated the questionnaire prior to testing on study participants. Please provide details regarding the validation group within the methods section.

Responses:

- The details of validating the questionnaires have added: “Piloting the questionnaire was implemented on 20 middle-aged women to ensure the logic, expression, understandability, and avoid confusion, misunderstanding to study participants”.

Position: 

Lines 105-107

3. In your Data Availability statement, you have not specified where the minimal data set underlying the results described in your manuscript can be found. PLOS defines a study's minimal data set as the underlying data used to reach the conclusions drawn in the manuscript and any additional data required to replicate the reported study findings in their entirety. All PLOS journals require that the minimal data set be made fully available. For more information about our data policy, please see http://journals.plos.org/plosone/s/data-availability. Upon re-submitting your revised manuscript, please upload your study’s minimal underlying data set as either Supporting Information files or to a stable, public repository and include the relevant URLs, DOIs, or accession numbers within your revised cover letter. For a list of acceptable repositories, please see http://journals.plos.org/plosone/s/data-availability#loc-recommended-repositories. Any potentially identifying patient information must be fully anonymized.

Responses:

- We have uploaded our study’s minimal underlying data set as either Supporting Information files during re-submitting. The file named “dataset.dta”

II/ Reviewer #1 Comments:

1. How were the income quintiles defined in terms of the unit of measurement? Can you provide a range of these units in each quintile (i.e. Lowest (0-10,000 USD))?

Responses:

We added the range and units of each quintile. We used "xtile" of Stata to calculate the income quintiles. Additional information in Table 1 & 3 includes:

"Income quintiles (million Vietnam Dong): 

Lowest (0.5-2)

Lower (2.5-4)

Medium (4.2-5)

High (5.3-12)

Higher (13-17.2)”

Position: 

Table 1 &3

2. Is the data available? If not, can you provide an explanation of its absence.

Responses:

- We have uploaded our study’s minimal underlying data set as either Supporting Information files during re-submitting. The file named “dataset.dta”

3. While the authors mention a few of the significant factors in table 2, they seem to miss discussing other significant variables. Can you please discuss table 2 a bit more in terms of other significant variables? Also, can you discuss the importance of the Total row in table 2?

Responses:

- Table 2 described the univariates which consist of domains on participant’s quality of life. These univariates play as outcome variates for the multivariate regression model. Moreover, our study purpose aimed to explain the association between QoL and independent variables of participants. Therefore, our discussion only focused on the significant results in Table 4 instead of discussing the descriptive results 

- The additional discussion has added in paragraph 2, as follow: “This result was lower than the previous mean MENQoL score which was explored in United Arab Emirates (3.03 - 3.61) [24], Saudi Arabia (2.28-3.19) [25], and China (2.33-2.84) [26] with sample sizes were respectively 70, 90 and 413. Furthermore, the present study indicated the lowest mean score in the psychosocial domain, the finding was similar to the prior report [24]. In contrast, the sexual domains showed the highest mean score, which was consistent with the Chinese women group [26]”

Position: 

Lines 184-189

4. I appreciate the information table 3 provides, however, the discussion of the table seems unclear.

Responses:

Table 3 provides demographic characteristics; therefore, we have replaced the position in Tables 2 and 3. As mentioned above, our study purpose aimed to explain the association between QoL and independent variables of participants. Thus, discussing the univariates in this table has been not suitable for study aims. 

5. Table 4 seems to provide coefficient values, however, the authors discuss odds ratios in their discussion of this table. Please choose either the coefficient values or ORs when reporting regressions

Responses:

-We have amended as follow: 

“Factors associated with quality of life among Vietnamese women were shown in Table 4. Older age was related to higher MENQoL scores in all four domains. People living in the rural areas had a positive relationship with the score of sexual domains (Coef. = 1.40; 95%CI: 0.78; 2.01). Additionally, patients' health issues such as migraine, diabetes had higher scores in vasomotor, psychosocial, physical domains. Outpatient was positively associated with MENQoL in domains of psychosocial, physical, sexual. 

In contrast, people being white-collar workers and businesses had lower scores of psychosocial and physical domains, while other occupations had higher scores of vasomotor and sexual domains. In addition, with higher entertainment active hours per day related to higher scores of all four domain, including vasomotor (Coef. = -1.51; 95%CI: -2.08; -0.93), psychosocial (Coef. = -0.96; 95%CI: -1.24; -0.68), physical (Coef. = -0.71; 95%CI: -0.86; -0.56), and sexual domains (Coef. = -1.19; 95%CI: -1.70; -0.69), similarly with doing exercise had positively related to the sexual domain (Coef. = -1.81; 95%CI: -2.52; -1.10). Participant was higher perceived social support score had negatively associated with vasomotor (Coef. = -0.54; 95%CI: -0.89; -0.18) and psychosocial domains (Coef. = -0.30; 95%CI: -0.48; -0.12).”

Position: 

Lines 165-175

6. I am unsure of why a Tobit regression was used because the authors do not explain the presence of censored data. Can you please explain the importance of this model in your manuscript?

Responses:

- We adjusted and added the information of censored data in the statistical analysis section, as follows: “Descriptive statistics were used to assess all of the questionnaire's items. The score of each item was converted ranging from 1 to 8 for analysis. 1 point is equivalent to participants answering ‘No’, while the scores from 2 to 8 correspond to the seven-point Likert scale (0-6). The variables of socioeconomic characteristics, daily activity, and perceived social support were considered as covariates, while the MENQOL (four domains) were served outcome variable.”

- The Tobit regression model is a frequently used tool for modeling censored variables in econometrics research. Previous studies were demonstrated that in the presence of a ceiling effect, if the conditional distribution of the measure of health status had uniform variance, then the coefficient estimates from the Tobit model have superior performance compared with estimates from other regression.

Position: 

The lines 134-139

7. The reporting of the multivariate model is unconventional and should include results for each domain unless there is a valid reason for the absence of these results.

Responses:

We had reviewed our multivariate model and recognized that it had already included the results for each domain.

8. Please expound on the main results in the discussion. I also recommend including literature from the SWAN study and the Midlife Women’s Health Study for comparison in the discussion.

Responses:

- The main results in the discussion had added on Paragraph 1 as follows: “Age, living in rural areas, and the existence of such health issues as migraine and diabetes were positively related to increasing the symptoms of perimenopause. On a notable point, findings showed that women’s quality of life in perimenopause might be improved by increasing the exercises, entertainment activities as well as perceived social support.”

Position: 

Lines 180-183

Responses:

- The SWAN study and the Midlife Women’s Health Study also were added for comparison in the discussion: “This result was lower than the previous mean MENQoL score which was explored in United Arab Emirates (3.03 - 3.61) [24], Saudi Arabia (2.28-3.19) [25], and China (2.33-2.84) [26] with sample sizes were respectively 70, 90 and 413. Furthermore, the present study indicated the lowest mean score in the psychosocial domain, the finding was similar to the prior report [24].”

Position: 

Lines 184-187

9. There are major grammatical errors needing to be addressed, I suggest language editing of the manuscript.

Responses: We reviewed and revised the grammatical errors and language editing of the manuscript.

III/ Reviewer #2 Comments:

1. Much of the needed methods description is missing. How many interviewers were involved, and how were they trained? Did they use a guided interview form for socioeconomic and activity data? Were interviews recorded and transcribed/scored later, or were they scored at the time of the interview? How were participants recruited? Please follow the STROBE guidelines (https://www.equator-network.org/reporting-guidelines/strobe/).

Responses:

The methods were added as follows: “Four medical volunteers at ward health centers were recruited and involved in the current study. Selected interviewers were able to handle forms and keep track of paperwork. They were trained in face-to-face interviewing skills, using questionnaires, and locating the sampled study subjects at household. An interview guideline form was developed for collecting data and scoring the measure scales in questionnaires.”

Position: 

Line 96 to 100

2. You are making causal claims with your language in the discussion, when your data are only correlational. Be careful about what you can conclude from a cross-sectional study design, and rewrite the description of your findings in terms of association only.

Responses:

The main findings in the discussion were amended: “Age, living in rural areas, and the existence of such health issues as migraine and diabetes were positively related to increasing the symptoms of perimenopause. On a notable point, findings showed that women’s quality of life in perimenopause might be improved by increasing the exercises, entertainment activities as well as perceived social support.”

Position: 

Lines 180-183

3. How much of the effect of exercise and leisure activity could be confounded by income and socio-economic status effects?

Responses:

Due to the current study question concentrated on the factors that affected the quality of life of perimenopause (4 domains), the variables included in the regression model just explore the relationship between MENQoL and covariates (income and socio-economic status ), without checking the association between characteristics factors (income and socio-economic status) and effect of exercise and leisure activity .... Therefore, we did not have enough data to discuss this point further.

4. There are quite a few typos, and no line numbers have been provided. Please edit carefully and add line numbers for ease of review.

Responses:

We had added the lines in the manuscript and amended typos.

5. Specific comments:

p. 11: suggest using “Asian” instead of “oriental”

Table 1: What were the income quintile values? Also, can you add the MENQOL data here?

Responses:

- “oriental” was replaced by “Asian”

Position: 

Line 71

Responses:

- Additional information in Table 1 & 3 includes:

"Income quintiles (million Vietnam Dong): 

Lowest (0.5-2)

Lower (2.5-4)

Medium (4.2-5)

High (5.3-12)

Higher (13-17.2)”

Position: 

Table 1 & 3

---

## [Decision Letter · Decision Letter 1]

25 Apr 2022

Physical activity and social support are associated with quality of life in middle-aged women

PONE-D-21-21709R1

Dear Dr. Nguyen,

We’re pleased to inform you that your manuscript has been judged scientifically suitable for publication and will be formally accepted for publication once it meets all outstanding technical requirements.

Kind regards,

Rebecca Lee Smith, D.V.M., M.S., Ph.D.

Academic Editor

PLOS ONE

Additional Editor Comments (optional):

Reviewers' comments:

Reviewer's Responses to Questions

**Comments to the Author**

1. If the authors have adequately addressed your comments raised in a previous round of review and you feel that this manuscript is now acceptable for publication, you may indicate that here to bypass the “Comments to the Author” section, enter your conflict of interest statement in the “Confidential to Editor” section, and submit your "Accept" recommendation.

Reviewer #1: All comments have been addressed

Reviewer #2: All comments have been addressed

2. Is the manuscript technically sound, and do the data support the conclusions?

Reviewer #1: Yes

Reviewer #2: Yes

3. Has the statistical analysis been performed appropriately and rigorously? 

Reviewer #1: Yes

Reviewer #2: Yes

4. Have the authors made all data underlying the findings in their manuscript fully available?

Reviewer #1: Yes

Reviewer #2: Yes

5. Is the manuscript presented in an intelligible fashion and written in standard English?

Reviewer #1: Yes

Reviewer #2: Yes

6. Review Comments to the Author

Reviewer #1: (No Response)

Reviewer #2: I would still be interested in follow-up analysis of cross-tabulation of the factors that were included in the multivariable model, but this is sufficient.

7. PLOS authors have the option to publish the peer review history of their article (what does this mean?). If published, this will include your full peer review and any attached files.

Reviewer #1: No

Reviewer #2: No

---

## [Editor Report · Acceptance letter]

28 Apr 2022

PONE-D-21-21709R1 

Physical activity and social support are associated with quality of life in middle-aged women 

Dear Dr. Nguyen:

I'm pleased to inform you that your manuscript has been deemed suitable for publication in PLOS ONE. Congratulations! Your manuscript is now with our production department. 

Kind regards, 

on behalf of

Dr. Rebecca Lee Smith 

Academic Editor

PLOS ONE